# Pharmacy Students’ Perceptions and Stigma Surrounding Naloxone Use in Patients with Opioid Use Disorder: A Mixed Methods Evaluation

**DOI:** 10.3390/pharmacy8040205

**Published:** 2020-11-03

**Authors:** Alina Cernasev, Michael P. Veve, Taylor Talbott, Elizabeth A. Hall, Kenneth C. Hohmeier

**Affiliations:** Department of Clinical Pharmacy and Translational Science, College of Pharmacy, The University of Tennessee Health Science Center, Nashville, TN 37211, USA; acernase@uthsc.edu (A.C.); ttalbot1@uthsc.edu (T.T.); liz.hall@uthsc.edu (E.A.H.); khohmeie@uthsc.edu (K.C.H.)

**Keywords:** naloxone, opioid use disorder, opioid stigma, addiction, pharmacy student

## Abstract

Pharmacists represent a key group of healthcare professionals that can increase awareness and destigmatize naloxone use. The objective of this study was to investigate pharmacy student perceptions of the use, dispensing, and stigma surrounding naloxone. An electronic survey was administered to pharmacy students that included questions about demographics, work history, naloxone use, and naloxone stigma. Separate qualitative interviews were performed to identify themes surrounding naloxone use. Two-hundred sixty-two participants completed the survey. The majority of participants were “highly willing” (74%) to fill a naloxone prescription for a patient and “somewhat comfortable” (38%) in counseling on naloxone; most were “somewhat comfortable” (38%) administering naloxone. Naloxone is “very rarely” (87%) recommended in community workplace settings, and the majority (64%) reported that patients never request information about naloxone availability. Seventy-six percent of respondents reported that naloxone-associated interactions have an influence on the way they communicate with patients in community pharmacy settings. Thematic analyses found that pharmacy students identify the importance of naloxone as a life-saving medication and the need for naloxone training, but patient-perceived stigma and limited access to naloxone remain prevalent. Pharmacy students are generally well-versed and inclined toward distributing, counseling on, and administering naloxone. Naloxone is rarely dispensed and patient conversations involving naloxone are infrequent in community settings. Future efforts focused on approaches toward difficult patient conversations and normalization of naloxone are needed to destigmatize and facilitate use.

## 1. Introduction

Over the past two decades, the United States (U.S.) has experienced an evolving public health crisis: the opioid epidemic. What started as an effort to provide adequate pain management transformed into widespread opioid overuse, overdose-related deaths, and other significant detrimental societal effects [1]. In 2000, prescribers increased their use of opioids as a result of Joint Commission pain management standards; pharmaceutical companies subsequently began heavily promoting their highly addictive opioid products [1]. From 1999 to 2018, almost 450,000 people died from a prescription or illicit opioid overdose [2]. While the introduction of naloxone products into the U.S. market and in pharmacies has helped to curtail the prevalence of opioid-related overdoses across the country, several barriers remain in obtaining access to naloxone and decreasing the negative stigma surrounding its use, from both a population and healthcare provider perspective.

Naloxone is an opioid antagonist medication that was Food and Drug Administration (FDA) approved in 1971 to reverse respiratory depression caused by opioid overdose [3]. The development of various naloxone products has quickly escalated in recent years as a result of the continuing epidemic of opioid-related fatalities [3]. Evzio^®^ is a naloxone auto-injector which was FDA approved in 2014, and Narcan^®^ is an intranasal naloxone spray approved in 2015 [4,5]. While traditional access to naloxone for opioid overdose reversal has been achieved through community-based education and distribution programs, recent legislative changes have seemingly increased the availability of naloxone in the community, with and without a prescription. Additionally, U.S. legislation has been designed to expand access to nearly 40 states to allow pharmacists to provide naloxone to patients and caregivers of those at risk of opioid overdose. This legislation exists in the form of collaborative practice agreements, standing orders, or granting prescriptive authority over naloxone to encourage dispensing [6]. Despite these changes, recent data still suggest that naloxone utilization remains limited [7,8,9,10].

There are several factors that are thought to contribute to the low acceptance of naloxone use in the U.S. Stigma towards individuals who use opioids, particularly from healthcare providers and law enforcement, may dissuade individuals with opioid use disorder (OUD) from seeking help due to the risk of judgment [7,8,9]. OUD-related stigma is thought to be related to a reflect a lack of understanding of addiction as a chronic relapsing brain disorder that can be effectively treated, which unfortunately further stigmatizes individuals with substance use disorders [10]. Accessibility of various naloxone products in pharmacies and product cost are also barriers to patient populations that generally lack access to care. These issues highlight the need for pharmacist educational efforts focused on naloxone access and recommendations for use in the appropriate populations. As highly accessible healthcare providers broadly distributed throughout communities, pharmacists are in an ideal position to promote safe opioid use and educate patients on the use of naloxone [11]. To further promote naloxone use and ensure a competent workforce, many U.S. Colleges of Pharmacy have implemented naloxone and substance use education into their curricula to equip them with tools to help combat the opioid epidemic [7,8,12]. There is, however, a lack of studies investigating pharmacy student perceptions of naloxone.

The objective of this study was to investigate pharmacy student perspectives of naloxone use, dispensing, and stigma. These results may be used as a guide for Colleges of Pharmacy to provide education and training on substance use disorder and appropriate naloxone use in the community.

## 2. Materials and Methods 

A mixed methods approach was used to examine the relationship between pharmacy students’ perspectives of stigma associated with naloxone use and dispensing [12]. An exploratory sequential design was employed in this study, where quantitative and qualitative data were collected sequentially in two phases [9,13]. In the first phase, quantitative data were collected and analyzed, while in the second phase, qualitative data were collected and analyzed to explain the findings of the quantitative study component [9]. The protocol for this study was approved by the University of Tennessee Institutional Review Board. Pharmacy students provided informed consent to participate in this study. Reporting of the mixed-methods approach in this study was guided and in agreement with the Good Reporting of a Mixed Methods Study (GRAMMS) criteria [14]. 

### 2.1. Quantitative Data

The quantitative component of this study was used to inform the qualitative portion in answering pertinent research questions and designing focus group questions. Students voluntarily completed an online survey focused on student understanding of naloxone usage and dispensing after completing an educational course. Students at the University of Tennessee Health Science Center (UTHSC) College of Pharmacy who were enrolled in a Doctor of Pharmacy program (*n* = 704) were eligible to participate. Survey respondents included students in their first (P1), second (P2), third (P3), and fourth (P4) professional year. The survey responses were captured and stored electronically using Qualtrics (Provo, UT, USA). The UTHSC Institutional Review Board granted exemption approval for this study. The survey was first distributed in December 2019 and responses were collated until March 2020. The survey invitation was sent two times within four weeks in between the initial email.

An 18-item survey was delivered electronically as previously published [15] in accordance to methodology derived from Dillman and colleagues [16]; seven closed-ended questions from this original survey item pertained to perceptions of naloxone use that have not been published previously (Appendix A). All data were aggregated such that no personal information linking specific responses to a specific participant was retained. The survey topics included questions related to naloxone availability at their workplace, as well as students’ beliefs and attitudes about naloxone prescription use, administration, counseling, and stigma associated with naloxone use. 

Participants were asked to indicate their level of willingness to fill naloxone prescriptions using a 5-point Likert Scale (1 = highly willing to 5 = very unwilling). Respondents with pharmacy experience were asked an additional two questions focused on participant comfort in administering naloxone and confidence in counseling a patient picking up a naloxone prescription, which was also assessed using a 5-point Likert Scale (1 = very comfortable to 5 = very uncomfortable and 1 = very confident to 5 = very unconfident, respectively). One additional item used a 3-point Likert scale (“very well”, “well”, “not well”, and “unsure”) to assess perceptions of how well the curriculum at the College of Pharmacy prepared students to recommend naloxone to patients at risk for or with OUD. 

An additional survey item assessed stigma manifestation, specifically if a student’s interactions towards a patient would change for if an individual was picking up a naloxone prescription, using a 5-point Likert scale (1 = strongly agree to 5 = strongly disagree). At the end of the survey, participants answered nine questions regarding demographics and pharmacy work history. Work history questions included the setting (i.e., community, hospital, or other pharmacy setting) and duration of employment. The survey participants could choose not to respond to any questions within the survey.

### 2.2. Qualitative Data

Recruitment for the focus groups occurred via email advertising; three focus groups were held until saturation of the data was achieved [17]. A total of 16 student pharmacists participated in the focus groups. The semi-structured focus group facilitator’s guide was developed by the qualitative researchers (A.C., K.C.H.) and practicing pharmacists (M.V., L.H.) and focused on different areas such as naloxone dispensing and the stigma associated with naloxone, and a scenario was provided to help them to better understand how to prepare future curricula. The facilitator’s guide used the theoretical elements from the Transtheoretical Model proposed by Prochaska, stigma concepts proposed by Link and Phelan, and Social Cognitive Theory proposed by Bandura [18,19,20]. The first and last authors (A.C., K.C.H.) led all focus groups, and both have past training and experience with qualitative data collection.

The facilitator’s guide used the theoretical elements from the Transtheoretical Model proposed by Prochaska, stigma concepts proposed by Link and Phelan, and Social Cognitive Theory proposed by Bandura [18,19,20]. 

### 2.3. Data Analysis

All survey data were analyzed using SPSS for MacIntosh, version 26.0 (IBM Corporation, Armonk, NY, USA). Descriptive statistics were calculated for all variables (i.e., median and interquartile range for nonparametric numeric data and frequencies and percentages for all nominal and ordinal data). Between-group differences between four student classes were determined using Kruskal–Wallis tests for all nonparametric numeric data for four-group comparisons. A subgroup analysis of responses targeted towards workplace naloxone distribution and counseling was performed on pharmacy students with community pharmacy experience to prevent biased responses. All tests were two-tailed, and an *a priori* alpha level of 0.05 was used to determine significance.

All the focus groups were audio-recorded, and both facilitators recorded field notes. Transcription of audio was performed by a professional service company and kept in a password-protected computer. When this study was completed, the electronic files were deleted, and the participants were not re-contacted. Thematic analysis of the data used the Braun and Clarke framework approach [21]. The research team followed the six steps recommended by Braun and Clarke of the data that aimed to capture associations between categories and extract and conceptualize the themes [21]. Two researchers independently read each transcript and coded inductively all the transcripts. The codes were clustered based on their similarities into categories [21]. The research team met multiple times to discuss the identified themes and to ensure that the codes and categories developed would capture all the data. The research team discussed the similarities and differences for each emergent theme. The process of thematic analysis was conducted using Dedoose (California, USA) qualitative software, which facilitates all the codes transcripts to be kept organized and allows comparison of frequency of codes across categories. 

## 3. Results

### 3.1. Quantitative Analysis

The overall survey response rate was 37% (262/704). Participant demographic characteristics are depicted in Table 1. The majority of survey respondents were third-year (P3) professional students with prior community pharmacy experience.

The majority of survey participants reported that the UTHSC College of Pharmacy prepares pharmacy students “well” in regard to recommending naloxone to patients (*n* = 124, 47.3%), as listed in Table 2. Most students reported being “highly willing” to fill a naloxone prescription for a patient, whereas the majority of students were “somewhat comfortable” administering naloxone to a patient. A very small proportion (<5%) of survey participants reported being uncomfortable with naloxone administration or were unwilling to fill a naloxone prescription. Between-groups analysis stratified by professional year of pharmacy school (P1, P2, P3, and P4 years) was performed, which found a significant difference in response to the query “How well do you think the curriculum at UTHSC College of Pharmacy prepares pharmacy students to be able to recommend naloxone to patients?” (Kruskal–Wallis mean rank score of 105.16 for P1s, 141.69 for P2s, 108.10 for P3s, and 115.02 for P4s; *p* = 0.003); there was a significant difference in response for P3 students when compared to P4 students (*p* = 0.006). There were no significant differences in survey responses between the different classes regarding comfort in administering naloxone to a patient (*p =* 0.90) or willingness to fill a naloxone prescription (*p* = 0.362).

Survey responses regarding naloxone dispensing and use in a pharmacy were then stratified for pharmacy students who reported that they worked as community pharmacy interns (*n* = 167); this was rationalized as these individuals represent a cohort of pharmacy students most likely to encounter patient–naloxone interactions (Table 3). Only 12.6% of survey participants reported that naloxone is regularly (i.e., daily or several times per week) recommended at their workplace, and the majority reported that patients never request information about naloxone availability (64.1%). While most survey respondents reported that they are confident in counseling a patient on naloxone, 76% of respondents reported that community-based naloxone-related interactions have an influence on the way they communicate with patients. 

Between-groups analysis was also stratified by professional year of pharmacy school (P1, P2, P3, and P4 years) for survey respondents who reported pharmacy experience. Only one significant difference was identified for the query “How confident do you feel when counseling patients on naloxone?” (Kruskal–Wallis mean rank score of 86.25 for P1s, 90.86 for P2s, 64.87 for P3s, and 75.13 for P4s; *p* = 0.019); there was a significant difference in response for P2 students when compared to P3 students (*p* = 0.017). No other significant differences were found for the remaining queries when stratified by professional year of pharmacy school (*p* > 0.05). 

### 3.2. Qualitative Themes 

There were three themes that emerged to characterize subjects’ perceptions of stigma surrounding naloxone, knowledge about naloxone dispensing, perceived impediments to naloxone dispensing:(1)Patient-perceived stigma and naloxone recommendations(2)Student perceptions of naloxone as a lifesaving medication(3)Pros and cons of the naloxone training OR subjects’ recommendations on the naloxone training

#### 3.2.1. Theme 1: Patient-Perceived Stigma and Naloxone Recommendations 

This theme uncovers the plausible origins of the patient-perceived stigma associated with naloxone from the pharmacy student perspective. Comments were related to the sheer difficulty of initiating conversations about naloxone with patients. Subjects described the situation as “awkward” because patients would take offense to the recommendation of naloxone co-dispensing with opioid medications. 


*“It is an awkward conversation. A lot of people do take offense to the conversation. So I think the best way to approach something like that would be to ask them what they know about naloxone, has their doctor talked to them about it, and then just kind of describe the purpose of it and you just have to kind of talk about it with a lot of like- you don’t want to sound judgmental in anything you’re saying. You don’t want to make them sound like an addict. These are all things that people think of right away, so you’re approach to it has to be very like logical. Like this is for your safety. This is for anyone’s safety who is on these medications. And just kind of getting rid of the message behind it that you’re an addict. You need naloxone, so everyone is different in how you can like- you might have to know the patient a little bit to know the best way to approach them. But a gentle approach is best.” (S11)*


Informants noted the need to be aware of patients’ self-perceived public stigma for having naloxone medication on-hand or in their homes.


*“I think another barrier a lot of these patients face is the negative implications [stigma] that these medicines carry today. Everything you see in the media or just out there in the world, in general, tends to be negative about these things, and that can kind of weigh on the patient, make them be a little hesitant towards asking questions about these medications or getting it.” (S12)*


#### 3.2.2. Theme 2: Student Perceptions of Naloxone as a Lifesaving Medication

Most of the subjects focused on the significance of counseling patients to have access to naloxone in critical moments. S6 describes with simple words why it is vital for patients who take opioid prescriptions to have access to the lifesaving medication.


*“I mean, accidental overdose […] I feel like naloxone- like it’s important, whoever is around them who needs to know because they’re going to be the ones who are going to recognize the signs and be able to dispense it. So not only is it like a peace of mind for the patient, it’s a peace of mind for the family.”*


Although all the subjects recognized the importance of recommending and encouraging the patients to have naloxone easily accessible, they also expressed their main concern regarding the cost of the medication as the main impediment for most patients. S8 presents this barrier in a generic way:


*“First barrier would be cost for a patient for naloxone.”*


S16 sees the unfamiliarity with naloxone as another impediment for patients to access it. Informants noted a gap between the pharmacy profession’s belief of naloxone as a life-saving medication and patients’ lack of awareness of the impact that it can have on reversing opioid overdose.


*“One of the barriers is awareness. I mean, we all know and talk about naloxone because we work with drugs. The word is not completely out on it. A lot of people wouldn’t know what you’re talking about when you say naloxone specifically. They might understand just a brand name, heard of it, not know what it is.” (S16)*


#### 3.2.3. Theme 3: Pros and Cons of the Naloxone Training OR Subjects’ Recommendations on the Naloxone Training

The following theme emerged as all of the participants responded to the scenario:


*“Let’s say when you are working, you notice the patient (whom you counseled last week) falls on the floor and displays the signs and symptoms of an opioid overdose. You recognize the patient as being Mr. XX who has been picking up opioid prescriptions from the pharmacy. Let’s say the pharmacist on duty gives you the naloxone to administer to Mr. XX. Could describe your comfort level to administer the naloxone that would save the life of Mr. XX?”*


The first part of the theme presents the subjects’ perspectives on the scenario, while the second part of the theme describes their recommendations on training within the pharmacy curriculum.

Most of the subjects expressed a comfortable level of administering naloxone in this hypothetical scenario. Students articulated a desire to have repetitive practice opportunities in naloxone administration to compliment didactic training. Such training was noted to facilitate comfortability using naloxone and potentially reduce anxiety surrounding its use in a real-world setting. S7 talks about how emotions might have an impact on her psychologically; however, her attitude toward the administration and saving a life would not prohibit her from administering the medication.


*“I think in that situation, comfort or not, you probably have that adrenaline rush too, and you might not be perfect because I’ve never administered it before, but I’m going to do everything I can to administer it the correct way.”*


Several of the subjects suggested that their comfort level depended on the naloxone formulation. S5 seems to hint at being “a little hesitant” with the IM formulation, saying,


*“I would feel very comfortable giving an intranasal naloxone considering the ease of administration. But if it was in the case of, say, IM naloxone, I think I would be a little hesitant because I’ve never been in a situation where I have to give an emergent medication like that.” (S5)*


S3 points out how the nature of the hypothetical scenario and a possible IM administration is more routine for him:


*“…I might be an exception because I’ve worked in emergency services, so that type of situation doesn’t freak me out maybe as much as the average person…” (S3)*


Although the subjects responded positively to the scenario, nine of them recommended that a refresher course online would be beneficial. Both S2 and S16 echoed the same concept. S2 says: 


*“Something online, and it doesn’t have to be mandatory or anything like that. It could just be sending an email saying, hey, here is a quick refresher if you need any help. Nothing discouraging…” (S2)*



*“Something like a video that would show us like, okay, this is how- this is what would potentially happen. Or a real-life scenario where, you know, we go up to the front of the room, somebody lies on the ground, and then we have to physically administer the naloxone to someone or us physically doing it because I think it goes in the same way of us getting our immunizations.” (S16)*


## 4. Discussion 

This mixed methods study found that pharmacy students are generally confident in counseling patients on naloxone, willing to dispense naloxone, and comfortable in administering naloxone to patients. However, our results suggest that naloxone is not commonly prescribed, recommended, or inquired about by patients in the community pharmacy setting. Major themes surrounding prevalent stigma and barriers to accessing naloxone were uncovered from qualitative analyses. These data suggest that greater access to and normalization of naloxone in community settings is needed. 

Survey respondents suggested that an educational course focused on naloxone use prepared students well or very well to make recommendations to patients. However, the majority of survey respondents reported that these conversations, and naloxone use in general, does not occur frequently in the community pharmacy setting. Survey and qualitative focus group participants also suggested there to be some hesitancy in administering naloxone in a life-saving scenario, especially related to the route of naloxone administration. Routine naloxone administration refresher courses and/or updates to currently available products should be provided to pharmacists and other healthcare professionals. 

Thematic analysis found that some students may struggle with these perceived difficult patient conversations that may include some awkwardness. Other colleges of pharmacy and healthcare professionals have explored naloxone education with positive results. A study from the Ohio State University College of Pharmacy found that a student-based naloxone and harm reduction educational pilot program provided value in learning about key naloxone counseling points, hands-on experience with demo training kits, and patient/pharmacist scenarios [22]. Other studies have investigated how medical, nursing, or pharmacy students’ knowledge levels changed throughout the course of the activity, as did their confidence in patient/provider situations [23,24,25]. 

A common theme throughout the present study was that naloxone inquiries and use are uncommon, which is thought to be due to low awareness, naloxone-associated stigma, and/or access to care. Patient unfamiliarity with naloxone may be related to pharmacist and pharmacy student difficulties in initiating appropriate conversations with patients, as outlined in this study’s qualitative analysis. Although published literature has discussed the life-saving nature of naloxone in overdose-related scenarios, the stigma associated with naloxone or opioid use disorder is not explored in any great depth from a pharmacy curriculum perspective. Naloxone-related stigma has been formally identified in medical professionals and students as preventing appropriate naloxone prescribing. A qualitative study of physicians and medical students found that physicians might feel uncomfortable prescribing naloxone at the risk of insulting a patient on lower doses of opioids, while potentially enabling opioid-addicted patients to continue to use it [26]. Inconsistencies in perspective vary across individuals prescribed opioids vs. those using illicit opioids; patients prescribed opioids never felt at risk of overdose, while those using illicit opioids felt empowered to carry naloxone and “be part of the change [27].” 

In general, expert opinion suggests that naloxone is only prescribed for those dependent on or addicted to opioids, which may deter patients from picking up naloxone prescriptions [28]. Other individuals may be concerned with judgement in inquiring about naloxone products in community pharmacies, or such that they would be labeled as addicts by law enforcement if found carrying it. One study found law enforcement to be the least likely to exhibit stigma on social media out of a group of potential naloxone providers [29,30]. A 2019 study on public perceptions of naloxone use found that 61% of survey respondents had never heard of naloxone or were unaware that a medication existed to treat opioid overdose. Of respondents that self-reported current or past opioid use, over half felt they were not at risk of overdose [28]. Contrary to these misconceptions, the range of patients that may benefit from a naloxone prescription includes any individual using opioids, especially at higher doses, those who use opioids in combination with other sedating medications such as benzodiazepines, and those taking opioids who suffer from a chronic medical condition such as the human immunodeficiency virus, liver or lung disease, in addition to those who are opioid-dependent or people who inject opioids [31]. 

The cost of naloxone remains a significant limitation to its use, particularly with the consideration that 30% or more of patients with OUD lack healthcare insurance [8,32]. For a product first introduced into the U.S. market in the 1970s, insurance and out-of-pocket costs associated with naloxone can vary significantly [33]. One study found that individuals at high risk for opioid-associated overdose do not receive naloxone, citing fluctuating cost as a potential barrier. The same study found that patients paid copays and had out-of-pocket costs for naloxone that were nearly triple the amount for the index opioid prescription that identified the patient as high risk [33]. One potential avenue to improve upon this issue is to develop strategies to identify high-risk patients for opioid overdoses and to pair these patients with naloxone prescriptions and counseling for use. These patients could then be captured through the electronic health record databases in hospital and community settings.

### Strengths and Limitations

While this study uses a unique methodology to better describe pharmacy student perspectives on naloxone, there are some notable limitations. This survey was administered to students enrolled at a single institution, which may limit the generalizability of the data. However, the UTHSC College of Pharmacy spans three unique geographic regions of Tennessee and is expected to provide a good representation of pharmacy student perspectives from various backgrounds in an area significantly impacted by the opioid epidemic. Part of this administered survey only solicited the opinions of students who were currently working in a community pharmacy setting. These perceptions may differ from other pharmacy students who practice in different settings; however, naloxone use is likely to be most pertinent in the community. Prior experience in a community pharmacy may have influenced perceptions of naloxone use and consequently participant survey responses, which is a limitation to this study. The survey method used in this study consisted solely of closed-ended questions, which may limit the exploration of beliefs and attitudes. During the spring semester of their first professional year, UTHSC pharmacy students learn how to administer naloxone. The significant difference in response to how well the curriculum prepares students to be able to recommend naloxone to patients that was observed may be attributed to the placement of naloxone-related sessions within the curriculum. Specifically, the curricular placement of naloxone administration may explain why second-year students responded to this survey item more favorably. The higher level of confidence in counseling patients on naloxone that was reported by second-year students may also be explained by the curricular placement of naloxone administration in the spring semester of the first professional year. Therefore, there may be some limitations to external validity when compared to where naloxone training is provided in other U.S. College of Pharmacy curricula.

## 5. Conclusions

The results of this mixed methods study found that pharmacy students display comfort, willingness to dispense, and confidence in administering naloxone to patients. Overall, a curriculum-based course focused on naloxone was found to be useful, whereas some students may be hesitant to administer naloxone in the field. A consistent finding of the survey and student interviews is that there is a low naloxone awareness for patients in the community, which likely leads to low uptake in at-risk individuals. Naloxone is also rarely dispensed in these settings, potentially due to the negative stigma surrounding opioid use and abuse. Still, a small portion of pharmacy students report that they change their behavior when responding to a patient encounter related to naloxone. These data suggest that additional efforts are needed to normalize OUD in the community as a treatable disease. Opportunities to address the normalization of OUD in the pharmacy curriculum could include the routine integration of substance use disorder-based cases throughout longitudinal therapeutics courses and through objective structured clinical examination (OSCE) practices. Early introduction to these concepts to doctor of pharmacy students, through cases and community-based OSCEs, could better prepare them to understand social determinants of health commonly observed in patients with OUD. Naloxone would ideally be offered to all patients deemed high-risk for opioid overdose when filling opioid prescriptions in community settings, but additional research is needed in order to target naloxone counseling to these high-risk individuals when patient data are limited. Increasing access to naloxone and/or medication-assisted therapies are also necessary to combat inherent stigma.

## Figures and Tables

**Table 1 pharmacy-08-00205-t001:** Demographic characteristics of pharmacy students responding to a naloxone-themed survey.

Characteristic, *n* (%) or Median (IQR)	Overall, *n* = 262
Age, years	24 (23–26)
Female sex	166 (63.3%)
Ethnicity	
Caucasian	174 (66.4%)
Asian	32 (12.2%)
African American	20 (7.6%)
Native American	1 (0.4%)
Prefer not to disclose	35 (13.3%)
Year in Pharmacy School	
First (P1)	31 (11.8%)
Second (P2)	81 (30.9%)
Third (P3)	78 (29.7%)
Fourth (P4)	50 (19.0%)
Did not answer	22 (8.4%)
Prior retail pharmacy experience	153 (58.4%)
Current pharmacy intern status	231 (88.2%)
Pharmacy Setting	
Community	167 (63.7%)
Hospital	67 (25.6%)
Other	28 (10.6%)
Pharmacy experience, years	1.5 (0.67–3.0)

**Table 2 pharmacy-08-00205-t002:** Pharmacy student survey responses regarding naloxone use and stigma, P1–P4 years.

Characteristic	Overall, *n* = 262
How well do you think the curriculum at UTHSC College of Pharmacy prepares pharmacy students to be able to recommend naloxone to patients? ^†^	
Very Well	77 (29.4%)
Well	124 (47.3%)
Not well	31 (11.8%)
Unsure	30 (11.5%)
How willing are you to fill naloxone prescriptions to patients?	
Highly willing	193 (73.7%)
Somewhat willing	34 (13.0%)
Neutral	32 (12.2%)
Somewhat willing	1 (0.4%)
Very unwilling	2 (0.8%)
How comfortable are you with administering naloxone to a patient?	
Very comfortable	78 (29.8%)
Somewhat comfortable	100 (38.2%)
Neutral	42 (16.0%)
Somewhat uncomfortable	30 (11.5%)
Very uncomfortable	12 (4.6%)

^†^ Differences in responses were noted when data were compared for students in P3 and P4 years.

**Table 3 pharmacy-08-00205-t003:** Pharmacy student survey responses regarding naloxone use and stigma of respondents that practice in community pharmacy settings.

Characteristic	Overall, *n* = 167
How often is naloxone recommended at your workplace?	
Daily	6 (3.6%)
Several times a week	15 (9.0%)
Once a week	24 (14.4%)
Less than once a week	78 (46.7%)
Never	44 (26.3%)
When working in a pharmacy, how often are you asked by your patients about naloxone availability?	
Weekly	6 (3.6%)
Monthly	16 (9.6%)
Quarterly	38 (22.8%)
Never	107 (64.1%)
How confident do you feel when counseling patients on naloxone? ^†^	
Very confident	58 (34.7%)
Somewhat confident	63 (37.7%)
Neutral	29 (17.4%)
Somewhat unconfident	10 (6.0%)
Very unconfident	7 (4.2%)
When a patient picks up a prescription for naloxone, it does not influence the way I interact with him/her.	
Strongly agree	53 (31.7%)
Agree	25 (15.0%)
Somewhat agree	49 (29.3%)
Disagree	23 (13.8%)
Strongly Disagree	5 (3.0%)
Did not respond	12 (7.2%)

^†^ Differences in responses were noted when data were compared for students in P2 and P4 years.

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
