# Peer review of "Pharmacy Students’ Perceptions and Stigma Surrounding Naloxone Use in Patients with Opioid Use Disorder: A Mixed Methods Evaluation"

_pharmacy, 2020, doi:10.3390/pharmacy8040205_

Round 1
Reviewer 1 Report
Abstract, line 23. Do not start a sentence with a number.
Page 2, line 53. registered trademark added to EVZIO? Same with Narcan line 54.
page 3. focus groups- how many students? how many individuals made up the expert panel and what qualified them as the expert? Who served as the facilitators for the focus groups?
Line 131 re refreshments does not add anything, please remove.
Table 1: your numbers for ethnicity add to 261, not 262. Please clarify this discrepancy.
Qualitative themes: while this section is very interesting it needs to be condensed. Can the quote be shortened? Placed in a table or in another format so they are not in the meat of the text.
Discussion: lines 335-336 seem redundant and should be removed
Conclusion: What would you suggest to improve things? What additional methods do you suggest to normalize OUD in the community?
Author Response
Comment 1. Abstract, line 23. Do not start a sentence with a number.
Response 1. We have made this change as suggested (line 23).
Comment 2. Page 2, line 53. registered trademark added to EVZIO? Same with Narcan line 54.
Response 2. We have added registered trademarks as suggested (lines 53-54)
Comment 3. page 3. focus groups- how many students? how many individuals made up the expert panel and what qualified them as the expert? Who served as the facilitators for the focus groups?
Response 3. Thank you for asking these clarifying questions. We have modified the text to answer these queries (lines 135-142):
“A total of 16 student pharmacists participated in the focus groups. The semi-structured focus group facilitator's guide was developed by the qualitative researchers (AC, KCH) and practicing pharmacists (MV, LH), and focused on different areas such as naloxone dispensing, the stigma associated with naloxone, and a scenario was provided to better understand how to prepare future curriculum. The facilitator’s guide used the theoretical elements from the Transtheoretical Model proposed by Prochaska, stigma concepts proposed by Link and Phelan, and Social Cognitive Theory proposed by Bandura [17,19,20]. The first and last authors (AC, KCH) led all focus groups, and both have past training and experience with qualitative data collection”.
Comment 4. Line 131 re refreshments does not add anything, please remove.
Response 4. We have removed this comment as suggested.
Comment 5. Table 1: your numbers for ethnicity add to 261, not 262. Please clarify this discrepancy.
Response 5. Thank you for pointing out this discrepancy. We have added one student who identifies as a Native American and updated Table 1 accordingly.
Comment 6. Qualitative themes: while this section is very interesting it needs to be condensed. Can the quote be shortened? Placed in a table or in another format so they are not in the meat of the text.
Response 6. Thank you for your suggestions. We have attempted to truncate the qualitative data section(s). However, condensing and cutting the student quotes would potentially remove or skew the meaning of the data. Furthermore, we feel that removing the quotes and adding them into a table will make the manuscript harder to read. Per Pharmacy publishing policy, tables or figures (and in this case quotes) should be included in the manuscript near their pertinent sections. In light of reviewer 2 and 3 recommendations, we have also decided to keep this part as it is due to the potential impact the removal of this data could have on the manuscript.
Comment 7. Discussion: lines 335-336 seem redundant and should be removed
Response 7. We have removed the redundant material as suggested.
Comment 8. Conclusion: What would you suggest to improve things? What additional methods do you suggest to normalize OUD in the community?
Response 8. We have expanded on our conclusions with possible alternative solutions to normal OUD in the community, and throughout pharmacy curriculum. Our suggested focus is to provide repeated examples of patient cases throughout PharmD coursework to better reflect the current climate of the opioid epidemic (lines 421-428).
“Opportunities to address the normalization of OUD in pharmacy curriculum could include the routine integration of substance use disorder-based cases throughout longitudinal therapeutics courses and through objective structured clinical examination (OSCE) practices. Early introduction to these concepts to doctor of pharmacy students, through cases and community-based OSCEs, could better prepare them to understand social determinants of health commonly observed in patients with OUD. Naloxone would ideally be offered to all patients deemed high-risk for opioid overdose when filling opioid prescriptions in community settings, but additional research is needed in order to target naloxone counseling to these high-risk individuals when patient data are limited.”
Reviewer 2 Report
Student Pharmacists’ Perceptions and Stigma Surrounding Naloxone Use in Patients with Opioid Use Disorder: A Mixed Methods Evaluation
student pharmacist -is it pharmacy students? Or pre-reg pharmacist ?
, but patient-perceived stigma and access to naloxone remain prevalent-this sentence is not clear
Seventeen-percent of respondents reported that naloxone-associated interactions have an influence on the way they interact with patients in community pharmacy settings.Question: do you mean drug interaction with naloxone?
I doubt the phase “normalization of naloxone” is suitable in this context. Normalization is the process of bringing or returning something to a normal condition or state.
Student pharmacists are generally well-versed and inclined toward distributing, counseling on, and administering naloxone. Question: Was this reported in this study?
The following findings does not seem to be align with the objective of the study:
Prior retail pharmacy experience 60%)
Year in Pharmacy School
|
First (P1) |
31 (12%) |
|
Second (P2) |
81 (31%) |
|
Third (P3) |
78 (30%) |
|
Fourth (P4) |
50 (19%) |
|
Did not answer |
22 (8%) |
Pharmacy Setting
|
Community |
167 (64%) |
|
Hospital |
67 (26%) |
|
Other |
28 (11%) |
Author Response
Comment 1. student pharmacist -is it pharmacy students? Or pre-reg pharmacist ?
Response 1. Thank you for identifying that we used two different nomenclatures to define our source population. For consistency, we have elected to use “pharmacy student” as the most appropriate description. These changes have been reflected throughout the manuscript.
Comment 2. , but patient-perceived stigma and access to naloxone remain prevalent-this sentence is not clear
Response 2. We have added the word “limited” in front of access to better reflect the intent of this sentence (line 31):
“…but patient-perceived stigma and limited access to naloxone remain prevalent.”
Comment 3. Seventeen-percent of respondents reported that naloxone-associated interactions have an influence on the way they interact with patients in community pharmacy settings. Question: do you mean drug interaction with naloxone?
Response 3. The intent of this question was to focus on how pharmacy students personally interact with patients who request or access naloxone. We have attempted to modify this sentence to be better reflective of its intent (lines 28-29):
“Seventeen-percent of respondents reported that naloxone-associated personal interactions have an influence on the way they communicate with patients in community pharmacy settings.”
Comment 4. I doubt the phase “normalization of naloxone” is suitable in this context. Normalization is the process of bringing or returning something to a normal condition or state.
Response 4. We respectfully disagree with the reviewer for this comment. In this context, normalization does refer to the procedure of decreasing negative stigma associated with naloxone use. To change the phrasing likely would detract from the impact of this statement, which is supported by our survey and interview findings. We would like to defer to the Editor if further changes are needed.
Comment 5. Student pharmacists are generally well-versed and inclined toward distributing, counseling on, and administering naloxone. Question: Was this reported in this study?
Response 5. Thank you for this comment. This was reported in the abstract (Lines 23-26):
“The majority of participants were “highly willing” (74%) to fill a naloxone prescription for a patient and “somewhat comfortable” (38%) in counseling on naloxone; most were “somewhat comfortable” (38%) administering naloxone)”
This data was also provided within the survey results (Lines 179-192) and Table 2.
Comment 6. The following findings does not seem to be align with the objective of the study:
Prior retail pharmacy experience 60%)
Year in Pharmacy School
|
First (P1) |
31 (12%) |
|
Second (P2) |
81 (31%) |
|
Third (P3) |
78 (30%) |
|
Fourth (P4) |
50 (19%) |
|
Did not answer |
22 (8%) |
Pharmacy Setting
|
Community |
167 (64%) |
|
Hospital |
67 (26%) |
|
Other |
28 (11%) |
Response 6. We feel that survey respondent demographics are important for the reader to better understand where these results are derived from. For example, other Colleges of Pharmacy may have naloxone education in different professional years, which could decrease the external validity of our results to others. The same could be said about the proportion of students in our survey with prior pharmacy experience; it is very likely that pharmacy students who practice in a hospital setting have little patient encounters in general that may detract from the validity of our results. We have also performed between group analyses to detect if there were differences in survey respondent results based on professional year of pharmacy school, as described in the results section.
Reviewer 3 Report
Dear Authors,
Many thanks for preparing this article and allowing me to review.
I think this tackles a very unique group of trainees who work mainly on the frontline of service delivery and have the best opportunity to offer, educate and supply naloxone to patients who are in the greatest need of the medication.
The paper is well written and presented but I had a few thoughts and comments:
Line 61 - recent data - can this be expanded on referenced on what the recent data is
Line 127 - could the survey be included as an appendix? (may help other educational institutes repeat the questionnaire with other groups of students from other professions or other countries)
Line 167 - Table 1 - I would be interested in seeing the breakdown of the years students are in allowing the discussion of experience (both as students and when working and training at university) and its impact on the completed responses. It would also allow a look at the return rate per year of in take to the course e.g. how big is the class and how many were completed by students in that year. Pharmacy setting also totalled up to 101%
Line 184 - Table 2 - again as above could this be split down per year so the effects of age and experience on the results could be explored further. Again "willing" and "comfortable" questions totalled at 101%
Line 190 - " These data" - This data
Line 196 - Table 3 can you please check totals again another 101%. Again a course year breakdown might show some light on the effects of age and experience. Could the comments from the paragraph starting on line 200 be included in the table (or as a separate table?)
Line 263 - Can I ask if the students / pharmacies regularly offer naloxone to patients when they collect their opioid prescription? Especially if the opioid is / appears to be a new medication for the patient. This is using any way the naloxone can be supplied.
Line 313 - Can I ask is Naloxone routinely offered? and Does the opinion of the patients match the opinion of the students or is it just a perceive judgement / feeling of the student?
Line 335 - Is naloxone regularly and routinely offered to all patients receiving prescriptions for opioids? if routine and offered to all this reduces and perceived stigma, and could be introduced as a "safety net"
One further comment / suggestion, would it be of value asking if previous experience supplying / using or witnessing an overdose, as this might have provide a different outlook from those individuals.
Once again many thanks for allowing me to review this and I think it is a very interesting insight
Best wishes
Author Response
Comment 1. Line 61 - recent data - can this be expanded on referenced on what the recent data is
Response 1. Thank you for this comment. We have added citations following this statement, that are elaborated on in the subsequent paragraph.
Comment 2. Line 127 - could the survey be included as an appendix? (may help other educational institutes repeat the questionnaire with other groups of students from other professions or other countries)
Response 2. We are grateful for this recommendation. We have added the survey as an appendix.
Comment 3. Line 167 - Table 1 - I would be interested in seeing the breakdown of the years students are in allowing the discussion of experience (both as students and when working and training at university) and its impact on the completed responses. It would also allow a look at the return rate per year of in take to the course e.g. how big is the class and how many were completed by students in that year. Pharmacy setting also totalled up to 101%
Response 3. I believe you are referring to the breakdown of students, by professional year, who were included in the qualitative interviews. Due to confidentiality concerns, no specific student data were collected for the purposes of qualitative interviews. This is necessary to preserve the integrity of the qualitative interview and analysis. However, we attempted to recruit students from all of the professional pharmacy student years to have a better representation of responses.
If you are referring just to the quantitative results presented in tables 2 and 3, we have attempted to clarify the impact of age and pharmacy experience through the results section, as outlined below in Responses 4-6.
We have also corrected the “pharmacy setting” totals to be more accurate.
Comment 4. Line 184 - Table 2 - again as above could this be split down per year so the effects of age and experience on the results could be explored further. Again "willing" and "comfortable" questions totalled at 101%
Response 4. Thank you for this comment. We performed between groups analysis to account for the differences mentioned (lines 184-192):
“Between-groups analysis stratified by professional year of pharmacy school (P1, P2, P3, and P4 years) were performed, which found a significant difference in response to the query “How well do you think the curriculum at UTHSC College of Pharmacy prepares pharmacy students to be able to recommend naloxone to patients?” (Kruskal-Wallis mean rank score of 105.16 for P1s, 141.69 for P2s, 108.10 for P3s, and 115.02 for P4s; P=0.003); there was a significant difference in response for P3 students when compared to P4 students (P=0.006). There were no significant differences in survey responses between the different classes regarding comfort in administering naloxone to a patient (P=0.90), or willingness to fill a naloxone prescription (P=0.362).”
There were some differences in response when comparing students earlier in the curriculum compared to later in the curriculum, as noted above. We feel there is no feasible way to include these data into Table 2 without creating confusion for the reader.
We have also adjusted the totals to add up to 100% in Tables 1, 2, and 3.
Comment 5. Line 190 - " These data" - This data
Response 5. Thank you, we have removed “these data” and instead referenced Table 3 in parenthesis (Line 200).
Comment 6. Line 196 - Table 3 can you please check totals again another 101%. Again a course year breakdown might show some light on the effects of age and experience.
Response 6. We have adjusted the totals to equal 100%.
Similar to our comments in Response 5, we did perform between groups analyses to account for age and experience in the manuscript text (lines 210-216):
“Between-groups analysis were also stratified by professional year of pharmacy school (P1, P2, P3, and P4 years) for survey respondents who reported pharmacy experience. Only one significant difference was identified for the query “How confident do you feel when counseling patients on naloxone?” (Kruskal-Wallis mean rank score of 86.25 for P1s, 90.86 for P2s, 64.87 for P3s, and 75.13 for P4s; P=0.019); there was a significant difference in response for P2 students when compared to P3 students (P=0.017). No other significant differences were found for the remaining queries when stratified by professional year of pharmacy school (P>0.05).”
Again, there was a slight difference in response when comparing students earlier in the curriculum compared to later in the curriculum.
Comment 7. Could the comments from the paragraph starting on line 200 be included in the table (or as a separate table?)
Response 7. We have amended Table 2 to include a footnote highlighting the differences in response rate between P3 and P4 pharmacy students. We have done the same for Table 3. Thank you for this suggestion.
Comment 8. Line 263 - Can I ask if the students / pharmacies regularly offer naloxone to patients when they collect their opioid prescription? Especially if the opioid is / appears to be a new medication for the patient. This is using any way the naloxone can be supplied.
Response 8. This was not captured in our survey; however it is very likely this is extremely variable in community pharmacies. Based on our survey results, naloxone is not recommended routinely by pharmacists, nor do patients inquire about naloxone. We feel this is an important take-home point of our research. We do think this should be a normalized approach in patients who are perceived to be at high risk for opioid overdose, however this information is limited in outpatient settings in certain cases.
Comment 9. Line 313 - Can I ask is Naloxone routinely offered? and Does the opinion of the patients match the opinion of the students or is it just a perceive judgement / feeling of the student?
Response 9. As discussed in Response 8 and from our survey results, it seems that naloxone is not routinely offered in community pharmacies. Unfortunately, we do not have access nor IRB approval to administer a survey to patients to further inquire about naloxone perceptions. However, we feel this is an extremely important point that we plan to explore with subsequent studies in high-risk populations for opioid overdose or abuse.
Comment 10. Line 335 - Is naloxone regularly and routinely offered to all patients receiving prescriptions for opioids? if routine and offered to all this reduces and perceived stigma, and could be introduced as a "safety net"
Response 10. As discussed in Responses 8 and 9, naloxone does not seem to be regularly and routinely offered to patients receiving opioid prescriptions. We agree that naloxone should be routinely offered to high-risk patients; we hope our data highlight the need for naloxone use to be normalized in community settings.
Comment 11. One further comment / suggestion, would it be of value asking if previous experience supplying / using or witnessing an overdose, as this might have provide a different outlook from those individuals.
Response 11. Thank you for this comment. We feel this could be a useful query to ask for a follow-up survey in the future.